# Experimental and Numerical Study of an Innovative Infill Web-Strips Steel Plate Shear Wall with Rigid Beam-to-Column Connections

**Wahab Abdul Ghafar [1,2], Zhong Tao [1,2,*], Yan Tao [1], Yingcheng He [1], Lei Wu [1] and Zhiqi Zhang [1]**

[1] Civil Engineering and Architecture Faculty, Kunming University of Science and Technology, Kunming 650000, China
[2] Yunnan Earthquake Engineering Research Institute, Kunming 650000, China
* Correspondence: taozhong@kust.edu.cn

**Abstract:** Steel plate shear walls (SPSWs) offer good energy dissipation capability when subjected to seismic forces as a robust lateral load resisting structure. This research investigated the cyclic behaviors of innovative infill web-strips (IWS-SPSW) and conventional unstiffened steel plate shear (USPSW) experimentally and numerically. As a result, two specimens of a 1:3 scale three-story single-bay IWS-SPSW and USPSW were fabricated and tested under cyclic lateral loading. Rigid moment-resistant connections were used for the steel plate shear wall beam-column connection. The steel shear walls with infill web strips showed high ductility and less shear load-bearing than the USPSW. The hysteresis results showed that the IWS-SPSW had high energy dissipation with no severe beam-columns damages. On the other hand, the USPSW displayed severe post-buckling, infill panel cracks, and first-floor column damages. Moreover, the IWS-SPSW shear strength did not fall in the test specimen beyond 2.5% average story drift, where the structure exhibited great seismic behavior. FE models were created and validated with experimental data. It has been proven that the infill web-strips can affect an SPSW system's high performance and overall energy dissipation. From a parametric study, the material features of the infill web-strips, such as steel strength and thickness, can enhance the system's impact even more.

**Keywords:** infill web-strips steel shear wall; failure mechanism; energy dissipation; hysteresis behavior; nonlinear FE analysis; parametric analysis

## 1. Introduction

In the early 1970s, engineers and researchers started to create systems and technologies that would allow a building or structure to respond to earthquake ground shaking without experiencing damage, securing the structure and facilities. Several well-known and very sound techniques can protect against seismic risks, such as creating a flexible foundation or base isolation system, vibrational control devices, and reinforcing the buildings by shear walls, moment-resisting frames, steel braced frames, and so on. Among all the mentioned methods, the steel plate shear walls (SPSWs) represent a revolutionary lateral load-resisting technique that could successfully brace a building against wind and earthquakes [1–7]. The structural system comprises steel plates that are one story high and one bay wide and connected to the surrounding beams and columns. The plates are put in one or more bays for the entire building height, resulting in a rigid cantilever wall that can withstand earthquakes. In addition, depending on the design approach, the surrounding steel frame may use either moment-resistant or direct beam-to-column connections [8]. The panels may be stiffened or unstiffened [9]. The SPSWs are suitable for either new construction or the seismic upgrading of an existing steel or concrete structure. The method is expected to be more cost-effective than concrete shear walls; for example, since foundation costs are lowered, the rentable floor area is enhanced, and the construction procedure involves

using only one trade on the job site. The construction of an SPSW core is relatively straightforward when unstiffened plates are employed. The SPSWs have fundamentally beneficial qualities for resisting seismically generated loads. Superior flexibility, robust resistance to degradation under cyclic loading, high initial stiffness, and, when moment-resisting beam-to-column connections are present, inherent redundancy and significant energy dissipation are exhibited here [10,11].

Furthermore, the SPSWs' low self-weight minimizes gravity and seismic loads delivered to the foundations, resulting in lower construction costs. The SPSWs stiffened to prevent out-of-plane buckling were used to design several existing SPSW buildings [12,13]. However, it has been proven that a significant increase in energy dissipated during lateral cyclic loading can be achieved by extensively stiffening the panel. In most cases, the costs involved will be prohibitive. It has previously been reported that buckling does not inherently represent the end of helpful behavior, and there is significant post-buckling resistance in an unstiffened shear panel. The load-resisting mechanism changes at the buckling point, transitioning from an in-plane shear field to an inclination tension field. When the panel is thin, buckling can occur at pretty low loads, and the panel's resistance is governed by the action of the tension field on the panel [9]. As a result, the diagonal tension field generates large axial forces and flexural moments in boundary elements (Beam, Columns); specifically, the design of columns in multistory buildings is complex [14]. Because of this vital issue, SPSWs aren't widely used. Researchers have proposed several strategies to alleviate the substantial demand for VBE (vertical boundary element) caused by diagonal tension in web plates, including light-gauge SPSWs [15,16], low yield point SPSWs [17–19], SPSWs with slit by [6,20], SPSWs with a partially connected web plate to columns [21,22], and self-centering steel plate shear wall with infill web-strips and solid web plates by [23], SPSWs with the considerable disconnected length of web plate to vertical boundary columns and SPSWs with peripheral circular holes [24–29]. Other studies [30,31], meanwhile, have adopted SPSWs with partial length connection and largely disconnected to vertical boundary elements. The experimental and numerical results proved that flexure and stiffness demands on vertical boundary elements could be achieved by reducing the connection length between infill web plates and vertical boundary columns. However, this approach has some weaknesses, such as losing VBE's capability to mobilize web panel shear strength and deterioration of the panel's ductile behavior due to web plate out-of-plane displacement beside the vertical free edges. Corrugated steel plate shear walls (CSPSWs) were also developed as an alternative to the traditional flat steel shear walls and experimentally and numerically studied by [25–27]. The CSPSW can significantly improve the out-of-plane and shear buckling loads of infill panels due to corrugation, but the ultimate stress is less [28,29].

This paper presents an innovative infill web-strips steel plate shear walls (IWS-SPSW) system. This system is composed of horizontal and vertical boundary elements, and the infill web strips are arranged uniformly to a condition in which the tension field's inclination angle is adjusted. The strips are connected to the boundary elements by a fin plate. The wider-length bi-diagonal strips are restrained together by a bolt connection to prevent significant out-of-plane deformation. Infill web strips have certain advantages over solid infill web plates, such as reducing the connectivity of web plates to boundary elements; therefore, this can produce less axial force and flexural moment to the boundary elements. Previous cyclic tests [32–34] have shown comparatively large cyclic strain concentrations at the corners where a gap between the horizontal and vertical fin plates caused the corners of the USPSW to fracture. Additionally, unstiffened SPSW ends up remaining relatively thin. The arrangement of large and thin steel plates during construction, particularly the field welding of the thin plates to the boundary columns and beams, can be challenging, and this new system can resolve these problems effectively. Based on the novelty of the IWS-SPSW system, this study aims to examine the mechanical properties of this proposed shear wall under a lateral cyclic loading test and numerical analysis. Two 1:3 scaled testing specimens, IWS-SPWS and USPSW, were designed and subjected to compare the seismic performance

and failure modes of the IWS-SPSW. Finite element models of the new shear wall were established using ABAQUS software, and the mechanical properties of the IWS-SPSW and USPSW were compared. Furthermore, a parametric study was conducted using ABAQUS software to evaluate the effects of the infill-strips thickness, the material yield strength, and the wider length of the bi-diagonal strip bolt connections. The following are the primary contributions of this study. (1) The new IWS-SSPSW configuration was proposed based on sustainable production, processing standardization, and maintenance ease. (2) The suggested shear wall's seismic performance was validated through lateral cyclic testing. (3) The suggested shear wall's FE models were built, and a series of parametric investigations were carried out to get insights into its mechanical performance.

## 2. Experimental Description

Under the cyclic loading protocol, this study investigated two test specimens, IWS-SPSW and USPSW. The IWS-SPSW and USPSW were 1:3 scale one-span third-story structures with rigid moment-resisting beam-to-column connections. First, a six-story steel structure composed of gravity frames and an SPSWs resisting system was designed, and the three bottom stories were taken for the experimental study. Figure 1 describes the design and analysis of SPSWs and test specimens.

### 2.1. The SPSW System's Design

The plan for the prototype was based on a regular six-story structural SPSW and steel-frame system, as seen in Figure 1a. Hence, the prototype incorporated a steel structure composed of steel-frame and the perimeter steel shear walls, with typically six stories. Each story height is 3.6 m. Dimensions of the structural plan are 24 m in the x-direction and 16 m in the y-direction.

Two steel plate shear walls are in each direction, with four SPSW on each floor. The configuration of the prototype construction included two gravity steel-frame bays in each of the four directions that encircled the perimeter. The steel-frame bays are 6 m, and the SPSW bays are uniformly 4 m from the center-to-center of the columns, as shown in Figure 1b. According to the ASCE-7-16 [30] and the available earthquake design parameters (Ss = 1.13 g short period ground acceleration, S1 = 0.53 g one-second period ground acceleration), the gravity dead-load of the floor was 4.0 kN/m$^2$, and the roof was 3 kN/m$^2$. The live load of the floor was 3.0 kN/m$^2$, and the roof was 2.0 kN/m$^2$. According to the proposed equivalent lateral force method equation, the seismic base shear was calculated using the ASCE-7-16 [30]. The seismic base shear force equals *VE = 0.101 W*, where *W* represents the effective earthquake weight, and *Cs = 0.101* represents the seismic response factor. Based on the minimum design loads and associated criteria for buildings and other structures, ASCE/SEI 7-16, the following critical combination load was used: *1.2DL + 1.6LL, 1.2DL + 1.0LL + 1.0E.*

The AISC 341 [29] design provisions code was utilized to design the gravity frames and SPSWs, and the design assumption is that the SPSW should resist 100% of the base shear force, and the remaining steel frames should bear the gravity loads. Structural steel material Q235 with the yield strength of 235 MPa and Q345 grade with the yield strength of 345 MPa were utilized for infill web plates and beam columns, respectively. The structural H-Section and HB-section profiles produced the beam and column cross-sections. AISC Seismic Standards [29] and AISC Design Guide 20 [1] recommend designing SPSWs with a preliminary equal braced modeling method. The design of the beam column elements and thickness of the infill plates are the primary considerations in this approach. In this method, the steel infill web plates are first substituted by diagonal tension strips (equivalent truss model) in the story. Following the design of the diagonal tension, corresponding trusses are transformed into infill web plates. Equation (1) was recommended by the Canadian Standards Association (CSA S16-19/2014) [35] to determine the steel panel thickness at the story.

$$t_{wi} = \frac{2A_i sin\theta_i sin2\theta_i}{L sin^2 2\alpha_i} \tag{1}$$

where $A_i$ denotes the area of the equals tension trusses and $\theta$ shows the inclination angle (calculated as the ratio to a vertical axis) of the equivalent trusses member at story $i$. $L$ is the distance between vertical boundary element centerlines.

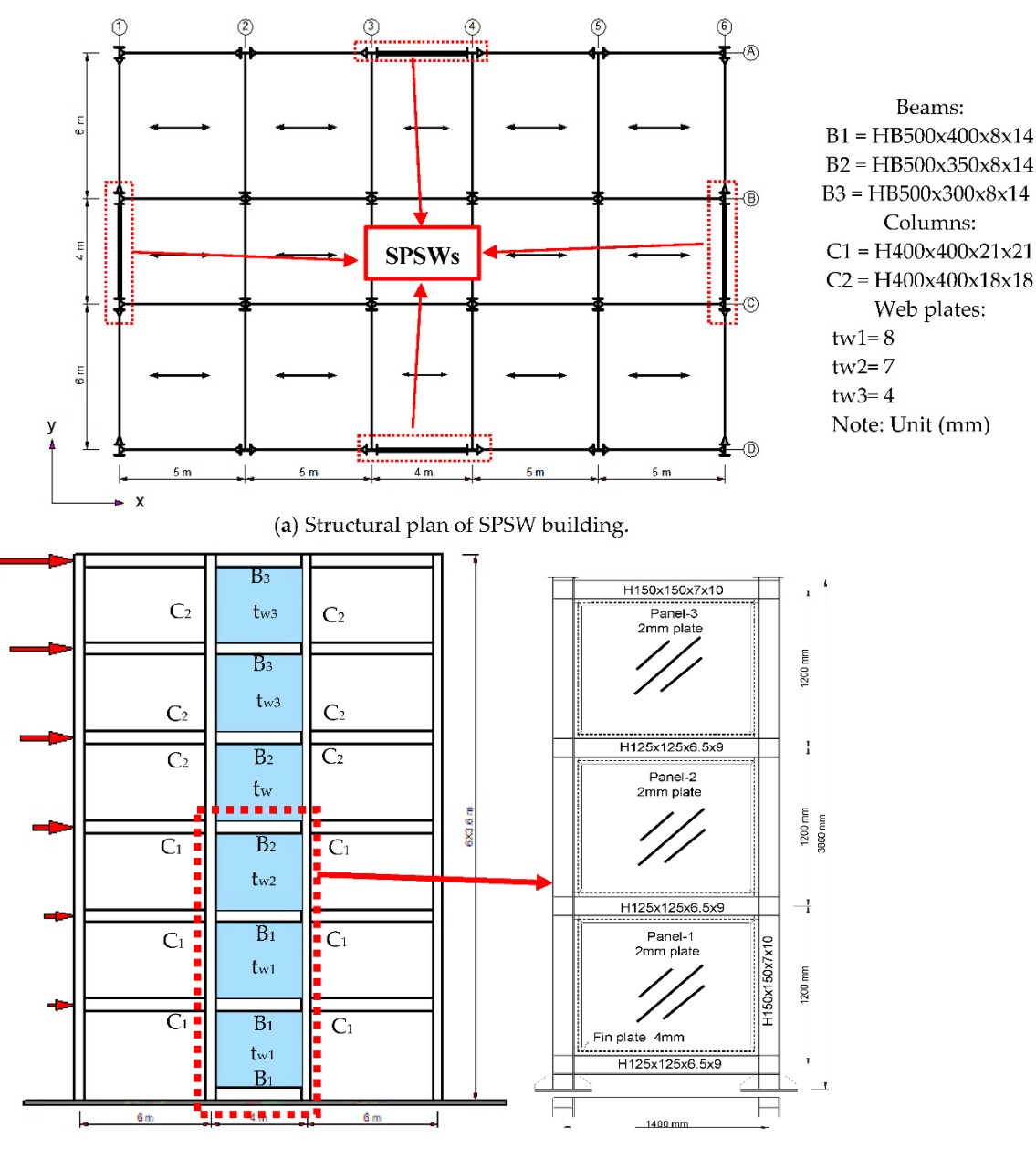

(**a**) Structural plan of SPSW building.

(**b**) Steel-frame and SPSW elevation          (**c**) scaled specimen

**Figure 1.** Details of the SPSW building and test specimen.

Equation (2) was introduced by AISC Seismic Standards [29], AISC Design Guide 20 [1], (CSA S16-19/2014) [35] to calculate the tension stress angle. Additionally, the angle of tension stress $\alpha$ was given between 30 and 50° for the preliminary design of web plate thickness.

$$tan^4\alpha = \frac{1 + \frac{t_w L}{2Ac}}{1 + t_w h \left[ \frac{1}{Ab} + \frac{h^3}{360 I c L} \right]} \qquad (2)$$

where $I_c$ denotes the moment of inertia of columns. $A_b$, $A_c$ indicates the beam and cross-sectional column area, and $h$ represents the height of the infill web plate.

For the infill plate to develop inelastic deformations, the boundary frame members must be rigid along their axis. Based on this, the AISC 341, AISC Design Guide 20, and the CSA S16-19/2014 [1] suggested that Equation (3) ensure that the columns have sufficient rigidity during the early design of steel shear walls. In Equation (3), $\omega_h$ denotes the column flexibility value.

The columns were designed using Equations (3)–(5): in the preliminary design stage.

$$\omega_h = 0.70h \left[ \frac{t_w}{2LI_c} \right]^{0.25} \tag{3}$$

$$\omega_h \leq 2.5 \tag{4}$$

$$I_c \geq \frac{0.00307 t_w h^4}{L} \tag{5}$$

The designing of the infill web plates, column, beam, and intermediary boundary components and the plate's widths to thickness ratio meet with the AISC 341 [36]. AISC and FEMA 450 [30,31] estimate the design shear stress of the infill webs plate, $V_n$, using Equation (6). Additionally, the columns and beams were constructed to make it conceivable to build the entire tension field action of the infill plate. According to the recommendations of AISC 341 Seismic Provisions, the columns were designed with the proper bending and shear capacities to resist the tension field created by the infill web plate. Equations (7) and (8).

$$V_u = \phi V_n = 0.42 f_y t_w L_{cf} sin2\alpha \tag{6}$$

$$M_u = \frac{1}{12} R_y f_y t_w h_c^2 sin^2\alpha \tag{7}$$

$$V_u = \frac{1}{2} R_y f_y t_w h_c sin^2\alpha + \sum \frac{0.5 M_{pb}^*}{h_c} \tag{8}$$

where $L_{cf}$ denotes the distance between column centerlines and $\phi$ is the resistance parameters provided (0.9) in AISC 341. $h_c$ represents the clear height of beams. $R_y$ is the anticipated stress at yield ratio to the provided minimum yield stress ($f_y$); $M_{pb}^*$ is the calculated plastic moment strength of the beams.

### 2.2. Prototype of Test Specimens

Two test specimens, 1:3 scaled three-story single-bay IWS-SPSW and USPSW, were examined using cyclic lateral loading (Figure 1c) and Figure 2. The details of beams, columns, and the infill web plate and infill-web strips of the experimental specimens are given in Table 1. The specifications for the test specimens include the material of the infill web plates, infill web strips, and the pattern of beam-to-column connections. In two test specimens, the thickness of the infill plate and infill web strips is 2 mm, and the infill web plates have an aspect ratio of 1.0. All vertical element members (columns) and top beam section H150 × 150 × 7 × 10 mm were used. On the other hand, section H125 × 125 × 6.5 × 9 mm was used for the middle beams.

**Table 1.** Test specimens.

| Specimen | Connection Type of Beam-Column | Plate Thickness (mm) | Middle Beam Sections | Top Beam Sections | Column Sections |
|---|---|---|---|---|---|
| IWS-SPSW | Moment-resistant | 2 | H125 × 125 × 6.5 × 9 | H150 × 150 × 7 × 10 | H150 × 150 × 7 × 10 |
| USPSW | Moment-resistant | 2 | H125 × 125 × 6.5 × 9 | H150 × 150 × 7 × 10 | H150 × 150 × 7 × 10 |

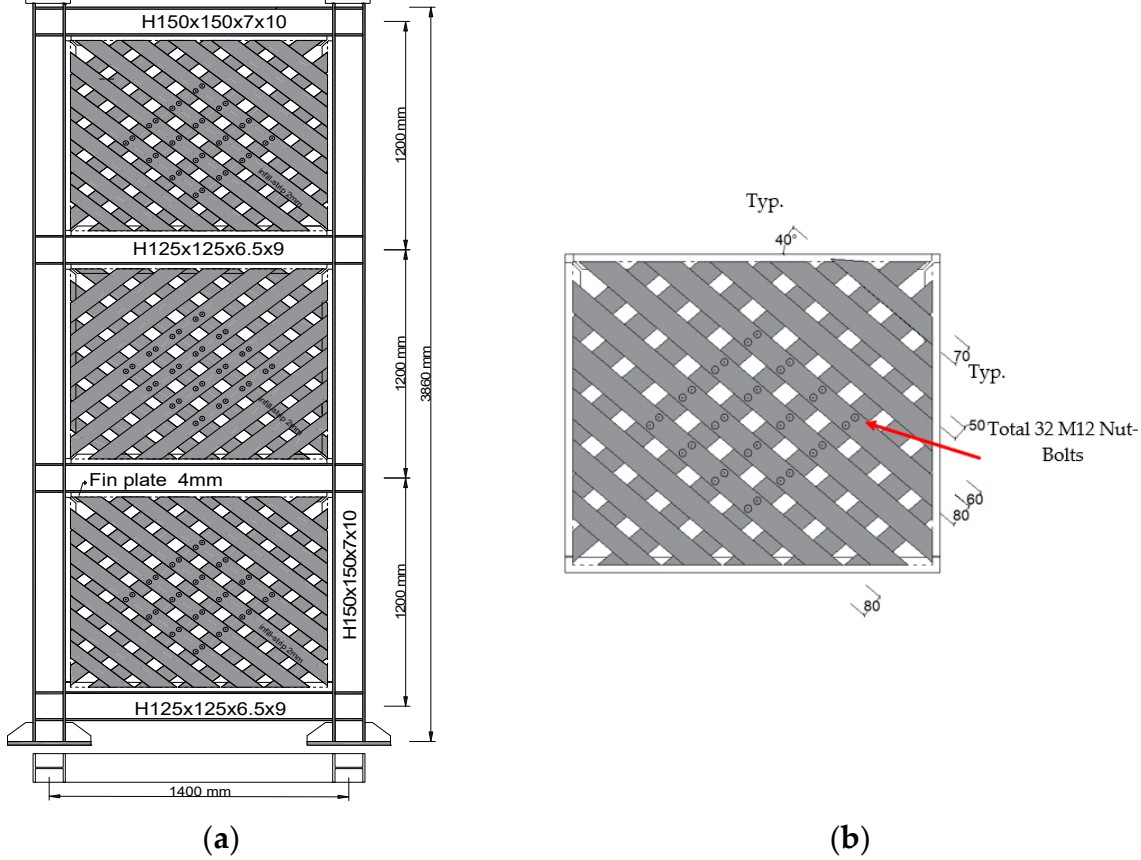

**Figure 2.** (**a**) IWS-SPSW test specimen (**b**) Infill web strip layout. Unit (mm).

Figures 1c and 2 represent the details of the IWS- SPSW and USPSW test specimens with rigid beam-to-column connections. Full-penetration groove welds connect the beam webs and flanges to the column flanges in the moment-resistant connection. In the IWS-SPSW test specimens, the infill web strips were placed on the opposite sides of the beam and column fin plates in right-leaning and left-leaning orientations, respectively. Figure 2a depicts the typical configuration of the IWS-SPSW and infill web strips. The distance between the strips was measured, and their spacing and length were changed to allow for openings. Therefore, an apparent size of approximately 50 mm and 60 mm was chosen for the test specimens. In addition, as can be seen in Figure 2b, the strip widths themselves vary from one another. Regular strips measure 70 mm wide, while wider strips measure 80 mm. Strips are provided near the beam-to-column joints and are wider to consider that a strip cannot be delivered in line with the connection joints. As seen in Figure 2b, the strips are pretty slender. Therefore, the broader strips are restrained using 12 mm diameter bolts to prevent significant out-of-plane buckling. The thickness of the infill web strips at each level in this testing program's specimens was identical to that of the infill web plate arrangement. In the USPSW, the infill plates were connected to the boundary frame members using a 4-mm thick steel plate as a fin plate. The boundary components were attached to the fin plates using double-sided fillet welding. Finally, the electrode type E43 was used for full-penetration grooves and fillet welding. Figure 3a,b shows the welding dimensions for attaching the fin plate to the boundary members.

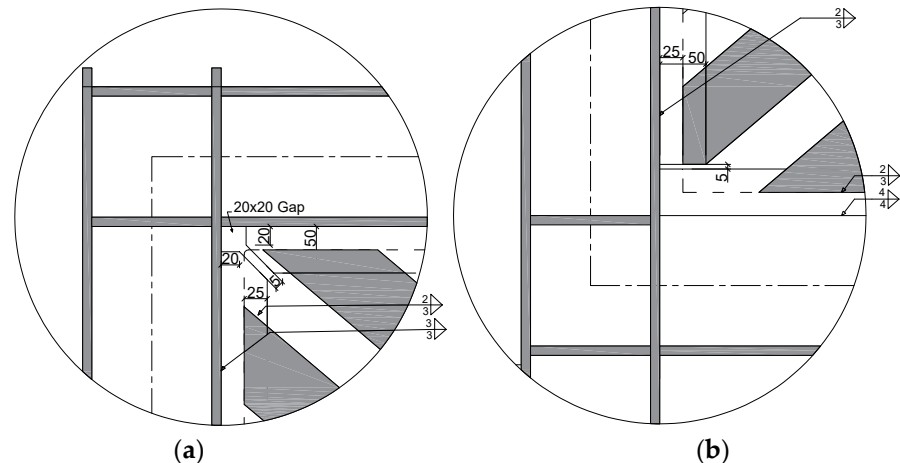

**Figure 3.** (**a**,**b**) Top and bottom floor of beam-columns, fin plate, and infill web-strips connections and details, respectively. Unit (mm).

### 2.3. Material Mechanical Properties

In order to create the experimental samples for the steel plate shear walls, two distinct types of steel were utilized. The steel (Q235) was used for the infill plate, and the steel (Q345) for the boundary element (beam-columns). Tests were performed per ASTM A370-17 [31] on three specimens of tensile test coupons for each thickness to assess the steel's material characteristics. Table 2 shows the coupon test results' yield and ultimate stress.

**Table 2.** The mechanical properties of steel coupon tests.

| Element | Steel Grade | Thickness (mm) | Modulus of Elasticity (GPa) | Yield Stress (MPa) | Ultimate Stress (MPa) | Elongation (%) |
|---|---|---|---|---|---|---|
| Web plate | Q235 | 2 | 200 | 242 | 370–460 | 21 |
| Beam & Columns | Q345 | 8.35 | 210 | 352 | 470–550 | 18 |
| Fin-plate | Q235 | 3.6 | 206 | 245 | 380–460 | 20 |

### 2.4. Test Setup and Instrumentation

The test setup, including the loading mechanism and boundary conditions, is depicted in Figure 4. The lateral loads were applied to one side of the top floor beam using a single hydraulic jack (Figure 4a–d). A total vertical load of 100 kN was given to a top beam, which was transferred to the adjacent columns. As indicated in Figure 4a, a lateral supporting frame was used to avoid out-of-plane distortion of the SPSW specimens. IWS-SPSW and USPSW specimens were tested at the Yunnan Earthquake Engineering Research Institute (YEERI). Displacements, forces, and stresses were measured using various equipment, such as a strain gauge, LC (load cell), LVDT (linear variable differential transformer), and cable potentiometer (CPs). The position of strain gauges, LVDTs, and LC measurements on the specimens is shown in Figure 4e,f.

### 2.5. Cyclic Loading Protocol

The top of each column was subjected to a 100 kN gravity force before any lateral loads were applied. This load remained consistent throughout the entire experiment. The experimental specimens were loaded laterally using a displacement control method applied in a cyclic quasi-static manner. For cyclic loading, the Applied Technology Council ATC-24's recommended loading history was used [32]. As demonstrated in Figure 5, the loading displacement history of the test specimens featured a step-by-step increment in the deformation cycles. Displacements began at 0.08% roof drift (3.096 mm) and subsequently

increased to 3.3% roof drift (116.1 mm). Story drift is calculated based on the maximum roof displacement divided by wall height ($=\delta_{max}/h$).

| (a) | (b) | (c) |
| --- | --- | --- |

| (d) | (e) | (f) |
| --- | --- | --- |

**Figure 4.** (**a**–**c**) photos of the tests, (**d**,**e**) Test setup, CPs, and LVDTs configurations, (**f**) strain gauge details.

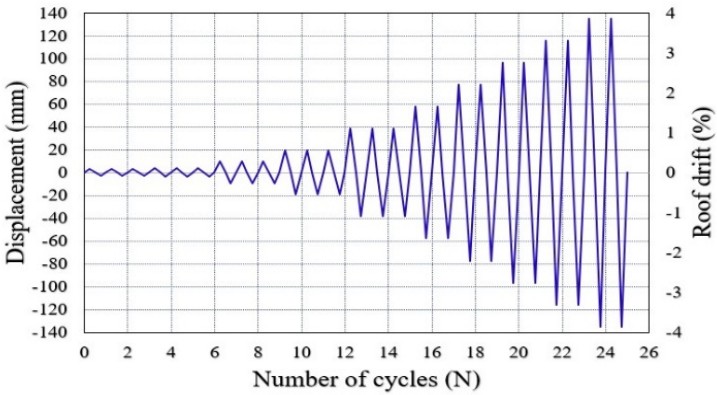

**Figure 5.** Applied load history.

## 3. Experimental Results

SPSW specimens were evaluated in quasi-static mode using the cyclic loading history from the preceding section. The findings of the SPSW experimental models are provided below. The observed impacts include cyclic behavior, hysteresis curve, ductility, stiffness, and energy dissipation capacity.

### 3.1. Cyclic Behavior of USPSW Test Specimen

During the first six loading cycles, an examination of the infill plate and boundary members revealed that no buckling or yielding occurred in the infill-web plate. In cycle 7, the infill plate initially yielded a horizontal displacement of 9.7 mm and a shear force of 82.35 kN. During cycles 8 and 9, the specimen responded similarly to cycle 7, and the yield developed on the infill plate. During these cycles, out-of-plane buckling occurred on the first and second floors (Figure 6a). Throughout loading cycles 10, 11, and 12, the yield of the infill plate continued to increase. Substantial deformation of the infill plates, the beginning of whitewash flaking, and column yielding were observed during the cycles of 13–15 (Figure 6b). The horizontal displacement and shear force in cycle 15 was 38.7 mm, and 349.25 kN, respectively. The first-floor left column's yielding began, and the whitewash flaking of the infill plates developed during loading cycles 16 and 17. The first-floor columns and infill plate yielded during loading cycles 18 and 19 with a horizontal displacement of 77.4 mm and shear force of 431.6 kN (Figure 6c). Minor tearing occurred in cycles 20 and 21 in the first-floor infill plate, displacing 96.75 mm and the maximum shear load of 437.1 kN (Figure 6d).

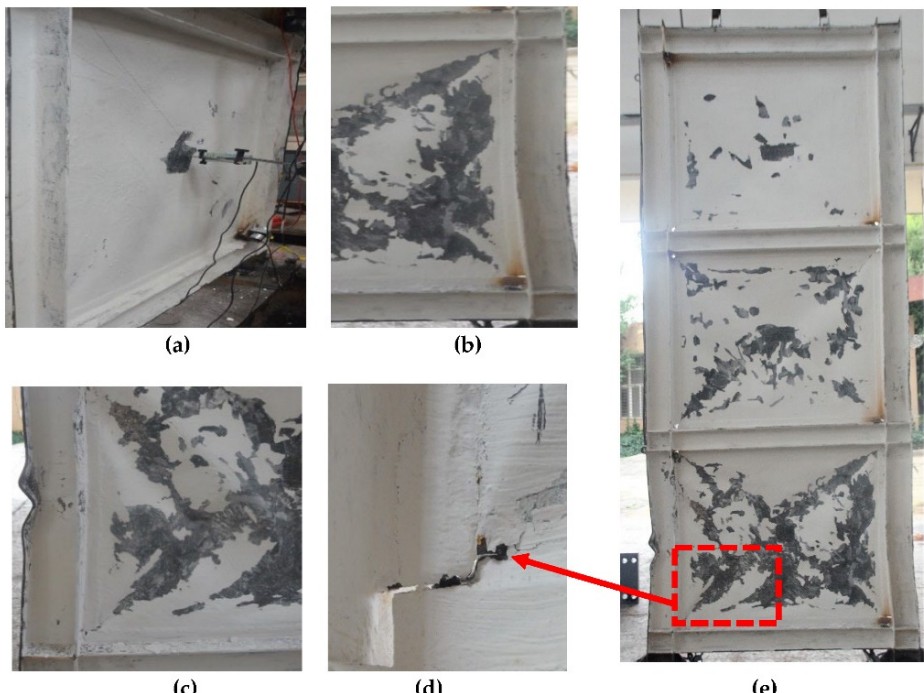

**Figure 6.** Details of USPSW specimen during the test: (**a**) first-floor buckling out-of-plane at cycle 7 (0.25% story drift), (**b**) fist-floor flaking of whitewash and column yielding (1.5% story drift), (**c**) first-floor column local buckling, (**d**) first-floor tearing corner of the infill plate (2.5% drift), (**e**) deformation of the specimen after the end of the test (3.3% story drift).

Additionally, during cycles 22 and 23, out-of-plane buckling and tearing of the infill plate occurred, as did the starting column flange bending on the first floor. The buckling intensity increased in all panels during the 21st cycle (Figure 6e). For example, the infill plate in the first-floor panel had an out-of-panel distortion of roughly 115 mm.

### 3.2. Cyclic Behavior of IWS-SPSW Test Specimen

During the first six loading cycles, evaluation of the infill strips revealed no buckling or yielding. Buckling and minor yielding occurred in the infill web-strip of the flooring during loading cycles 7, 8, and 9 (Figure 7)). In cycle 10, the infill web-strips initially yielded at a horizontal displacement of (9.675 mm), and the load was 82.2 kN. In cycle 10, minor paint flaking was detected on the infill strips near the junction of the horizontal and vertical fish plates on the first and second floors. The magnitude and area of the out-of-plane deformation infill plates increased on the second and third floors during cycles 11 and 12. In cycles 13–14 and 15, flaking increased in the top-left of the second floor's infill plate and the bottom-left of the third floor's infill plate (Figure 8b). Cycle 13 had a lateral displacement of 19.35 mm and a shear force of 162.5 kN, respectively. In cycle 15, an increase in flaking of the whitewash on the infill plate of floors was noted. During this cycle, the whitewash flaked to the extent that it covered the whole infill web strips. Moreover, the first-floor columns started to yield.

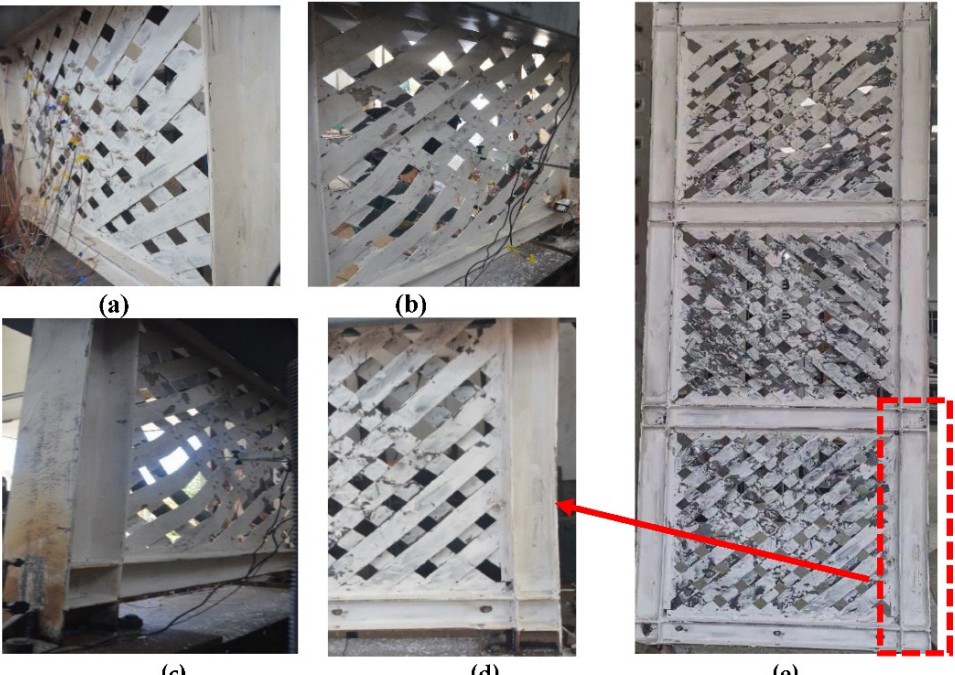

**Figure 7.** Details of IWS-SPSW specimen during the test: (**a,b**) first-floor buckling out-of-plane at cycle 7–19 (0.25–2% story drift), (**c,d**) yielding of the first-floor columns, (**e**) severe deformation of the specimen after the end of the test (3.3% story drift). Note: story drift is calculated based on the maximum roof displacement divided by wall height (=δ_max/h).

The first-floor column flanges, wrinkles, and the sign of plastic hinges were observed in cycles 16–17 (Figure 7c). The moment-resisting joints at the beam-to-column connections showed no yielding in the panel zones until the cycle19. During cycle 19, The load of 371.8 kN was recorded, corresponding to a maximum deflection of 77.4 mm. The maximum load of 381.2 kN was reached in cycle 22 with a horizontal roof displacement of 116.1 mm. This cycle resulted in the first-floor columns' complete yielding and yielding of the panel zones of the moment-resisting joints at the beam-to-column connections. No local or global buckling and rupture of the column and beams were detected during the test in specimen IWS-SPSW (Figure 7e).

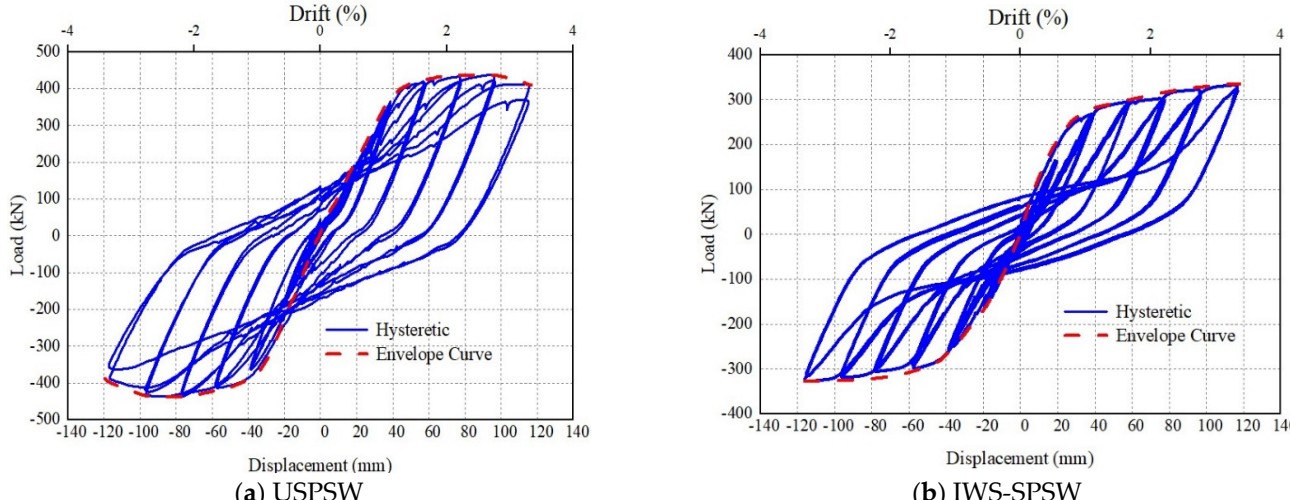

**Figure 8.** Hysteretic and envelope curves for test specimens.

### 3.3. Hysteresis and Envelope Curves Behavior of the Specimens

The hysteretic behavior of the specimens' IWS-SPSW, USPSW, and envelope curves are shown in Figure 8. Hysteresis loops exhibited essential characteristics, including strength, stiffness, ductility, dissipated energy, and the behavior of specimens observed in the tests. According to Figure 8, all parts of the infill plate participated in the energy dissipation of the system, and the model had centering and stable hysteresis loops. In the early (elastic) loading cycles, the panel behaves stiffly. In specimen USPSW, as the deformations increase, portions of the yield gradually reduce stiffness. After significant yielding of the infill panels has occurred, unloading and reloading in the opposite direction produce a consistent and characteristic hysteresis pattern. The maximum load achieved in each cycle increased slightly with each excursion to a new deflection level until the maximum base shear of 438.2 kN was reached in cycle 20. This took place at a deflection of 97 mm. Subsequently, the shear wall's load-carrying capacity declined gradually from cycle to cycle. Cycle 22 was also the cycle where panel tears first occurred. Tearing and the local buckling in the column flanges that began forming in cycle 22 have contributed to the gradual degradation of the specimen. In specimen IWS-SPSW, a high initial stiffness with less energy dissipation was evident in the elastic region. In the post-yield area, several well-defined segments of the load-deformation curves represented the various loading, unloading, and reversal of the infill web-strips buckles. Increased energy dissipation was achieved with each displacement level increment in the post-yield region. Due to local damage, a decrease in energy dissipation between subsequent cycles at the same load level. The principal sequence of significant inelastic action in the single-panel specimens yielded the infill web strips and yielding the boundary frame. The envelope curve of the specimens was obtained by successively connecting the peak points of the hysteresis curve of the first cycle at each stage in the same direction. The IWS-SPSW specimens exhibited a stable ductile behavior that increased the load-carrying capacity continuously (Figure 8). The USPSW specimens, on the other hand, exhibited a gradually decreasing load-carrying capacity after the maximum load-carrying capacity was reached during the early loading stage).

The failure modes in specimens USPSW and IWS-SPSW under cyclic loading in the hysteresis curve are exhibited in Figure 8. In the USPSW specimen, the infill plate buckles in the elastic zone, plate yielding under tensile stress, columns yielding, columns in-plane buckling, and tearing of the infill plates occurred, respectively. In the IWS-SPSW specimen, the infill web-strips buckle in the elastic area, the infill-plate yielding under tensile stress, and the columns yield, respectively. The fracture did not occur in the beam and columns in both specimens. The tearing happened at the edge of the infill plate on the first floor, which decreased USPSW specimens' capacity, and no fracture in specimen IWS-SPSW was observed.

### 3.4. Initial Stiffness and Stiffness Degradation

An idealized elastoplastic (IEP) curve and the load–displacement envelope curve of test specimen findings are shown in Figure 9. The IEP envelope curve is drawn using equal plastic energy [37,38]. The yield position is determined by comparing the area contained by the test envelope to the idealized elastoplastic envelope curve. Using the IEP curve, the initial stiffness of specimens ($K_{iy}$) is calculated as the force ratio to the elastic region's displacement. As a result, the initial stiffness of USPSW and IWS-SPSW specimens is 15.42 and 12.62 kN/mm, respectively, as presented in Table 3. For the stiffness degradation, the stiffness secant is used to describe the stiffness degradation of the test specimens, as illustrated in Figure 10. Form an envelope curve; the stiffness secant equals the gradient of the line passing between (0,0) and point ith ($\delta_i$, $P_i$); as displacement increases, specimen stiffness degrades, as shown in Figure 10.

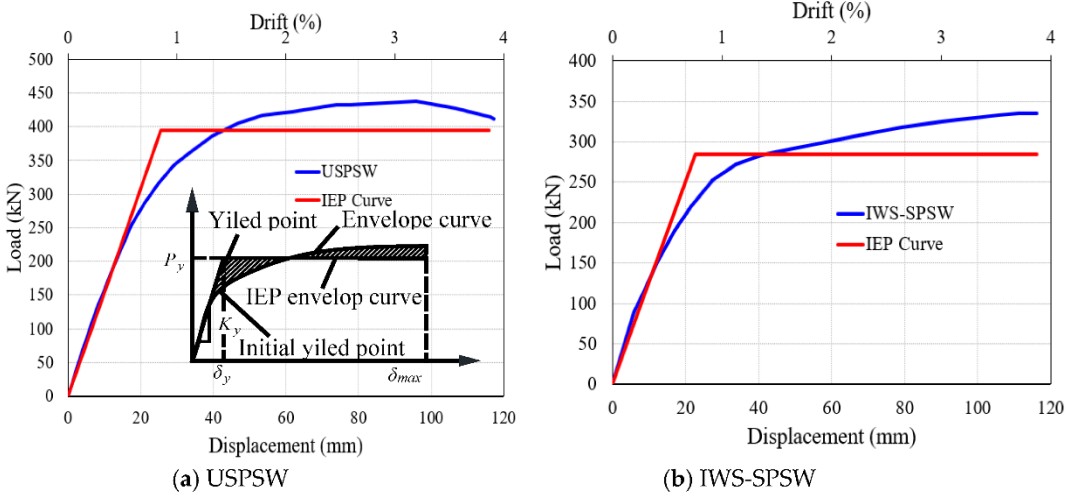

(**a**) USPSW        (**b**) IWS-SPSW

**Figure 9.** Envelope curve corresponding to IEP curve of test specimens.

**Table 3.** Test results.

| Specimen | Py (kN) | Pmax (kN) | K$_{iy}$ (kN/mm) | δy (mm)(drift%) | δmax (mm)(drift%) | Displ. Ductility |
|----------|---------|-----------|------------------|-----------------|-------------------|------------------|
| USPSW    | 394.05  | 437.8     | 15.41            | 25.55 (0.70)    | 116.1 (3.3)       | 4.54             |
| IWS-SPSW | 284.7   | 334.93    | 12.62            | 22.56 (0.61)    | 116.1 (3.3)       | 5.15             |

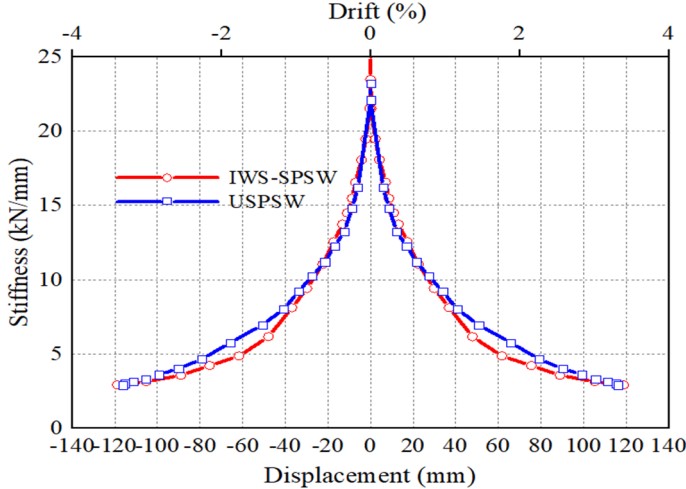

**Figure 10.** Stiffness degradation.

### 3.5. Displacement Ductility of the Specimens

Figure 9 shows the result of the test specimens for the envelope curve of the load-displacement and the idealized elastoplastic (IEP) curve. Equal plastic energy is used to depict the idealized elastoplastic envelope curve [7]. The yield position is determined so that the area bounded by the test envelope and the idealized elastoplastic envelope curve are equal. In assessing the structural design and seismic performance, the displacement ductility characteristic ($\mu$) is defined as the maximum relative lateral displacement ($\delta_{max}$) to the maximum relative lateral displacement at the yield point ($\delta_y$), or ($\mu = \delta_{max}/\delta_y$) sees (Figure 9a). Table 3 shows the specimens' ductility values as calculated. The ductility values for the USPSW and IWS-SPSW were 4.54 and 5.15, respectively, and the IWS-SPSW showed high ductility. The yield point ($\delta_y$) for both test specimens was calculated from the envelope curves, approximately 0.70% and 0.61 % drift (roof displacement = 25.55 and 22.56 mm). The specimen $P_y/P_{max}$ ratios are equivalent to 0.85, suggesting better ductility.

### 3.6. Energy Dissipation Capacity

The capacity for energy dissipation can be determined by calculating the region underneath the hysteresis loop [39]. Figure 11 depicts the area that is contained within a hysteresis loop. According to the loading cycle number, the energy dissipation of the test specimens is demonstrated in Figure 12. The energy dissipation capacity is modest during the first six loading cycles since the system is in the elastic phase. The IWS-SPSW specimen dissipated more energy than the USPSW until 1 % drift. The specimens' ability to dissipate energy increased as the number of loading cycles increased, as seen in Figure 12. In all loading conditions, the first-floor columns have a greater capacity to dissipate energy, particularly in the inelastic area.

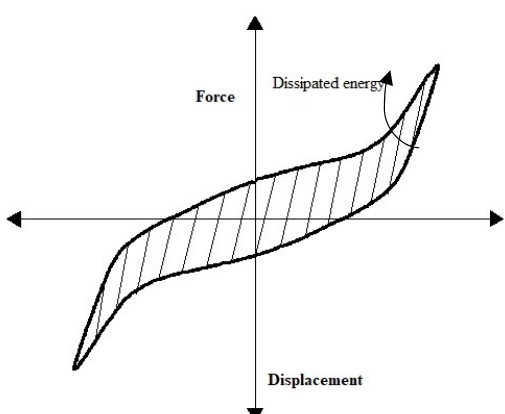

**Figure 11.** Calculative method of energy dissipation capacity of the hysteresis curve.

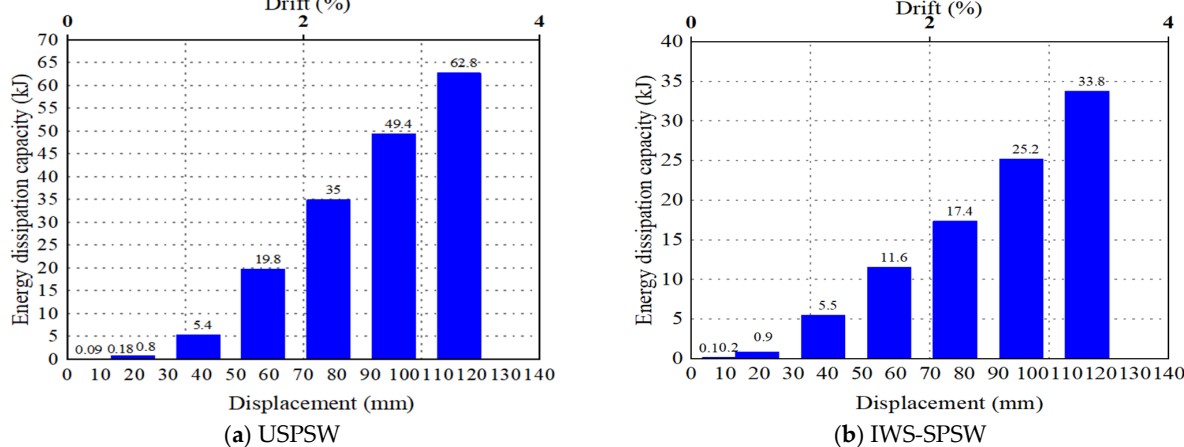

**Figure 12.** Energy dissipation capacity in the specimens.

In the USPSW specimen cycle 22nd, the maximum peak energy dissipation was 62.8 kJ, measuring more than twice the IWS-SPSW 33.8 kJ because of the damage contribution of the first-floor column.

## 4. Numerical Analysis

The nonlinear finite element models of the tested specimens were established by utilizing a commercial general-purpose nonlinear program, ABAQUS [40]. In order to study the behavior of the IWS-SPSWs, a three-dimensioned analysis is required to fulfill the infill panels and strips' out-of-plane deflections. Three-dimensional modeling is more complex because of the specialty of the infill strips. The details of FE models are presented below.

### 4.1. Material Properties

The cyclic constitutive model (combined hardening) is adopted to simulate the hysteretic behaviors accurately [25], which was proposed by Chaboche [37], consisting of a nonlinear kinematic hardening component and an isotropic hardening component. The parameters for this model are defined in ABAQUS (Hardening = Combined model). The hardening-combined constitutive model parameters can be obtained by data fitting [41,42]. The hardening-combined parameters of Q235 and Q345 steel are presented in Table 4.

**Table 4.** Combined isotropic and kinematic hardening in ABAQUS.

| Steel | fy/MPa | Q∞/MPa | Biso | C1/Mpa | $\delta 1$ | C2/Mpa | $\delta 2$ | C3/Mpa | $\delta 3$ | C4/Mpa | $\delta 4$ |
|-------|--------|--------|------|--------|-----|--------|-----|--------|-----|--------|-----|
| Q235 | 235 | 2100 | 1.2 | 7493 | 750 | 6273 | 514 | 2354 | 186 | 950 | 166 |
| Q345 | 345 | 1450 | 0.4 | 7994 | 650 | 6120 | 510 | 2265 | 176 | 875 | 158 |

### 4.2. Loading Procedure and Boundary Conditions

Based on the experimental data, FE models of SPSWs were subjected to boundary conditions, such as lateral support and cyclic lateral loading. In order to prevent the frame from moving out of the plane, the supports are moment-resistant and laterally supported, as shown in Figure 13. Practically, every panel has some manufacturing errors (i.e., they are not exactly flat). The plate's mode shapes were determined using an eigenvalue buckling analysis. The FE model was modified with minor out-of-plane deformations corresponding to the panel's lowest eigenmode (fundamental mode). The initial imperfection's magnitude was limited to $0.01 = \sqrt{l_p x h_p}$, where $lp$ and $hp$ reflect the plate's length and height, respectively [43], this perturbation forms the diagonal tension field in the panel. It mitigates the ideally flat plate's artificial over-stiffness. FE models used welded connections of restrained tie interactions to simulate the weld. The bolts connection of infill web-strips is defined based on defined mesh-independent fasteners [40]. As part of the SPSWs testing, the infill plate was connected to the boundary via the fin plate. In order to accomplish this goal, mesh-independent spot weld connection processes were implemented into the FE modeling of the fin plate.

### 4.3. Meshing and Analysis Methodology

The entire specimen was represented by the four-node, quadrilateral, stress/displacement shell elements using reduced integration and a large-strain formulation (ABAQUS S4R Element). The response of nonlinear geometries, strain hardening, substantial deflections, and post-buckling was considered in the finite element modeling of SPSWs. The nonlinear FE method can find the implicit and explicit solution techniques for physically challenging situations. The implicit method solution is best when working with static or quasi-static data [42,44–46]. The Newton–Raphson method is used to solve the nonlinear equations of FE models [40].

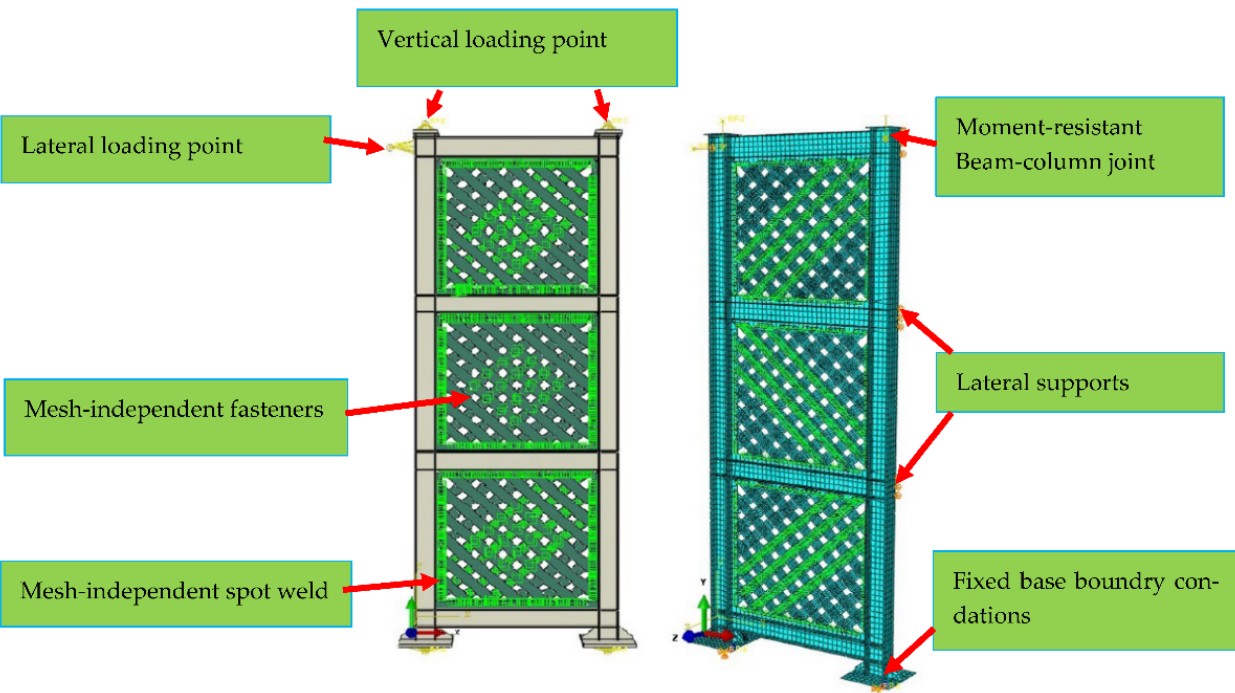

**Figure 13.** The developed FE model of the IWS-SPSW specimen.

## 5. Verification of FE Models

Using the ABAQUS software, the FE models of the SPSWs are created and checked for accuracy in this section [40]. A comparison was made for validation between the hysteresis and enveloped curves and the failure mechanisms of finite element models and experimental results. The equivalent plastic strain, also known as PEEQ [33], is considered to assess localized failure.

### 5.1. Hysteresis and Envelope Curve Comparisons

Figure 14 compares the hysteretic and envelope curves from the experiment and FE models. Due to a more optimal loading condition in FE analysis, the hysteretic curves produced by FE analysis are smoothly stiffer than those created by testing. The FE model's load-bearing capacity ratio and the test specimen were 1.05 and 0.99, respectively. The experimental results generally follow a pattern similar to the FE model.

### 5.2. Comparisons of Deformation and Failure Mode

The failure mechanism's areas can be estimated using the equivalent plastic strain (PEEQ) [40]. FE models and test specimen failure modes are depicted in Figure 15 using the equivalent plastic strain (PEEQ) concentration. Figure 15 illustrates failure modes of infill plates caused by a maximum PEEQ finite element model on test specimens, including plastic hinges on columns, out-of-plane deformation of infill plates, and infill plate corners tearing. PEEQ results revealed that the FE models accurately predicted the likelihood of failure mechanisms in cyclic loads.

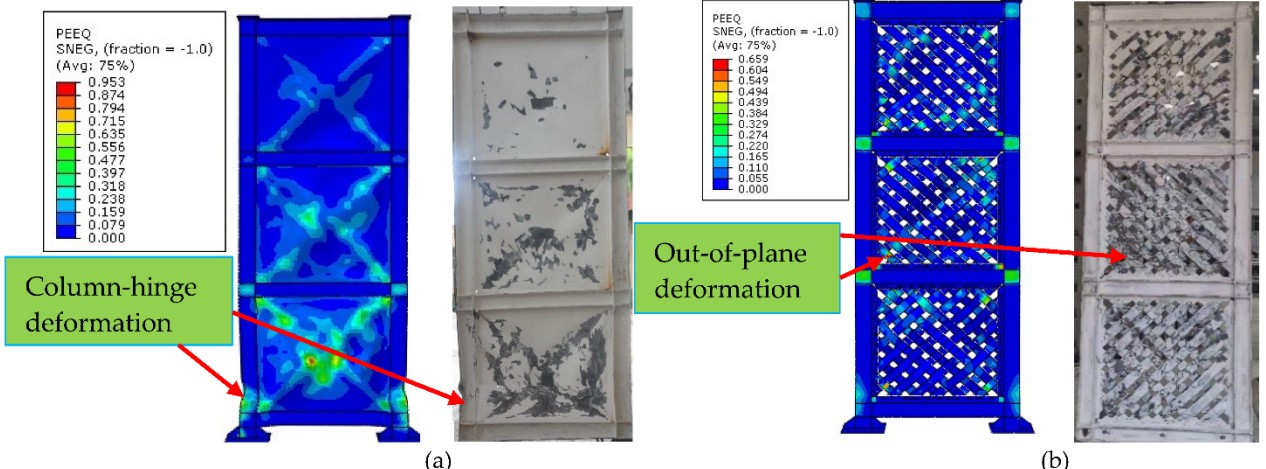

**Figure 14.** Comparison of hysteresis and envelope curves of tests and FE models: (**a**,**c**) USPSW, (**b**,**d**) IWS-SPSW.

**Figure 15.** Comparison of the test specimen and FE model failure modes: (**a**) USPSW and (**b**) IWS-SPSW.

## 6. Parametric Analysis

The parametric analysis focused on developing the numerical models was used to analyze the influence of the infill web-strips thickness and yield strength of the infill web strips on the efficiency of the steel shear wall systems. The parametric study generated and examined six three-story IWS-SPSWs models with varying configurations of strip thickness, infill web-strips connections, and infill web-strips yield strengths. The material parameters of the investigated IWS-SPSW models are summarized and described in Table 5.

**Table 5.** Parametric studies of the models and findings from the parametric analysis.

| Model | Material Property | | Strips Thickness (mm) | Infill-Strip Nut-Bolts | $K_{iy}$ (kN/mm) | $\delta_y$ (mm) | $\delta_{max}$ (mm) | $P_y$ (kN) | $P_{max}$ (kN) | μ | $E_D$ (kJ) |
|---|---|---|---|---|---|---|---|---|---|---|---|
| | Beam & Columns | Infill-Strips | | | | | | | | | |
| IWS-SPSW | Q345 | Q235 | 2 | yes [a] | 12.62 | 22.6 | 116.1 | 284.7 | 334.9 | 5.6 | 243.9 |
| IWS-1 | Q345 | Q235 | 2 | no | 16.76 | 21.7 | 118 | 241.2 | 331.1 | 5.4 | 232.7 |
| IWS-2 | Q345 | Q235 | 2 | yes [b] | 14.11 | 20.7 | 118 | 292.5 | 344.1 | 5.7 | 306.2 |
| IWS-3 | Q345 | Q345 | 2 | yes [a] | 14.31 | 23.5 | 118 | 335.9 | 395.3 | 5.0 | 261.8 |
| IWS-4 | Q345 | Q235 | 3 | yes [a] | 20.51 | 17.4 | 118 | 356.4 | 419.3 | 6.8 | 340.8 |
| IWS-5 | Q345 | Q235 | 4 | yes [a] | 24.32 | 18.2 | 118 | 442.9 | 521.1 | 6.5 | 405.9 |
| IWS-6 | Q345 | Q235 | 5 | yes [a] | 30.89 | 15.4 | 118 | 475.8 | 559.7 | 7.3 | 477.1 |

Note: [a] denotes as same as the test infill-strips nut-bolts connection shown in Figure 2b. [b] indicates that bolt connections connect all strips.

The displacement-controlled type simulation was used as the same specimen test cyclic loading procedures and a nonlinear pushover analysis. The findings of the parametric analysis of numerical models are provided in Table 5. Table 5 summarizes the initial stiffness, yield displacement, yield shear strength, maximum shear strength, displacement ductility, and cumulative energy dissipation capacity. Table 5 below contains an abbreviated form of the term for parametric models. (1) Infill web-strips steel plate shear wall without infill-strips nut-bolts connection (IWS-1). (2) Infill web-strips steel plate shear wall with all strips are prevented from significant elastic buckling by bolt connections (IWS-2). (3) Infill web-strips steel plate shear wall with the same boundary element and infill plate material (IWS-3). (4) Infill web-strips steel plate shear wall with infill strips thickness of 3.4 mm and 5 mm (IWS-4, IWS-5, and IWS-6), respectively.

### 6.1. Effect of The Infill-Strip Bolt Connections

A total of three models (IWS-SPSW, IWS-1, IWS-2) were considered, as detailed in Table 5. Cyclic and nonlinear pushover analysis were performed on all the models to study the effects of the infill web-strips bolt connection on the inelastic response of the system.

Figure 16a illustrates the von-mises stress response and post-buckling of test specimen (IWS-SPSW) with initial stiffness 12.32 (kN/mm) and 284.7, 334.93 kN yield and ultimate base shear as well as maximum strip deformation of 79.5 mm. Figure 16b, IWS-1 model was exhibited 16.76 (kN/mm) initial stiffness and 241.2, 331.10 kN yield, and ultimate base shear, respectively. At the loading end, the maximum strip deformation was measured 127.8 mm. Figure 16c represents the IWS-2 model with entire bi-diagonal strips bolt connections; the finding from the inelastic response showed an initial stiffness of 14.11(kN/mm) and 241.2, 331.10 kN yield, and ultimate base shear, respectively. The infill web-strips deformation was measured at the end of the loading analysis by 45.4 mm. The IWS-2 exhibited exemplary performance and high strength and ductility in comparing three FE models. Figure 17a–d represents each model's hysteresis behavior, pushover curves, and idealized elastic to plastic (IEP) curves.

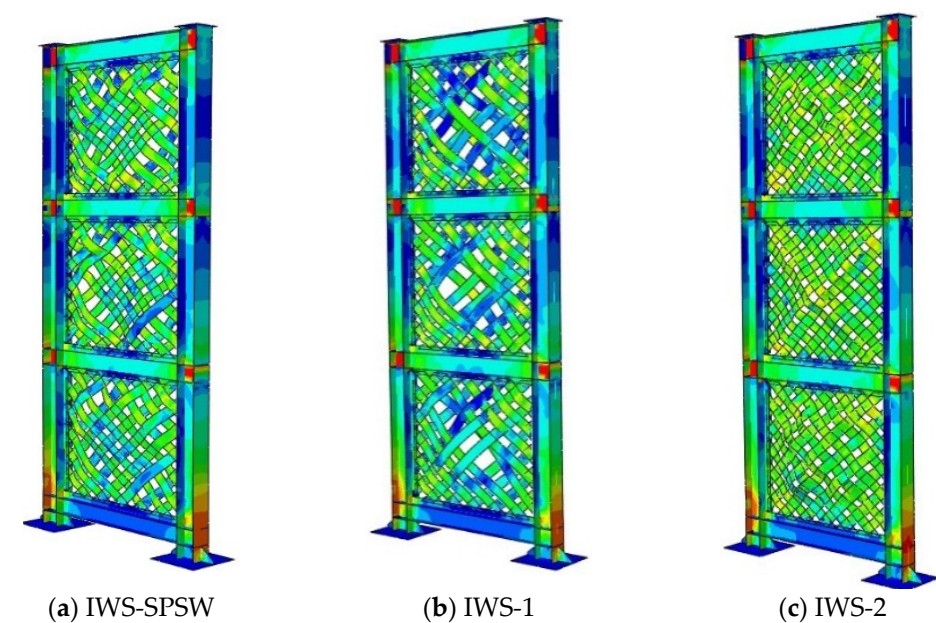

(**a**) IWS-SPSW         (**b**) IWS-1         (**c**) IWS-2

**Figure 16.** Parametric FE models: Effect of the infill-strip bolt connections.

(**a**) IWS-SPSW

(**b**) IWS-1

(**c**) IWS-2

(**d**) IWS-SPSW, IWS-1,2

**Figure 17.** Hysteresis, pushover, and IEP response of the FE models.

## 6.2. Effect of the Infill Web-Strips Yield Strength

Figure 18a indicates that the IWS-3 model with a high infill web-strips steel yield strength. The inelastic response of the model showed an initial stiffness of 14.31 (kN/mm) and 335.95, 395.24 kN yield and peak base shear, respectively. Compared to the previous models, the load-bearing capacity was significantly increased, and the energy dissipation was reduced. Figure 19a,e illustrate the parametric models' hysteresis, pushover, and IEP curve behavior.

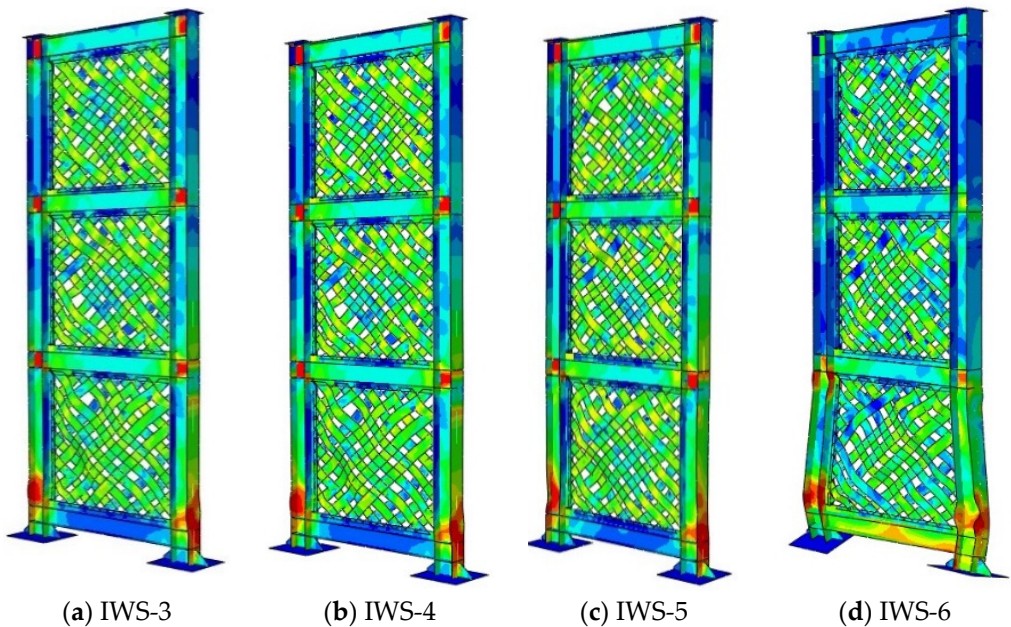

| (**a**) IWS-3 | (**b**) IWS-4 | (**c**) IWS-5 | (**d**) IWS-6 |

**Figure 18.** Parametric FE models: Effect of the steel yield stress and infill-strip thickness.

## 6.3. Effect of Infill Web-Strips Thickness

Figure 18b–d illustrates the model's von-mises stress response and hysteresis behavior of the three models (IWS-4-5-6) with the same infill web strips width, vertical and horizontal loading, and different infill web-strips plate thickness. The IWS-4 model with 3 mm infill web-strips thickness exhibited an initial stiffness of 20.51 (kN/mm) and 356.35, 419.25 kN yield, and ultimate base shear, respectively. The IWS-5 model with 4 mm infill web-strips thickness displayed an initial stiffness of 24.32 (kN/mm) and 442.89, 521.04 kN yield and ultimate base shear force, respectively. The IWS-5 model observed an in-plane failure at the first-floor columns. Therefore, the hysteresis and envelope curves showed a slight deterioration after the lateral drift of 2.5 (%). The IWS-6 model with 5 mm infill web-strips thickness showed an initial stiffness of 30.89 (kN/mm) and 475.76, 559.72 kN yield and ultimate base shear load. In the IWS-6, an out-of-plane failure occurred after the 1.8 (%) lateral drift, which caused a sharp deterioration of the hysteresis and envelope curves.

The IWS-4 model has exposed an excellent load-bearing capacity without any significant beam-column in-plane or out-of-plane deformation. On the other hand, the IWS-5 has displayed the best performance with a considerable in-plane of first-floor column failure. Finally, IWS-6 has shown remarkable performance in load-carrying capacity with more energy dissipation. It was revealed that by increasing the thickness of infill strips, the boundary elements experienced critical axial forces. The hysteresis loops, pushover, and IEP curves of IWS-3,4,5,6 models are shown in Figure 19.

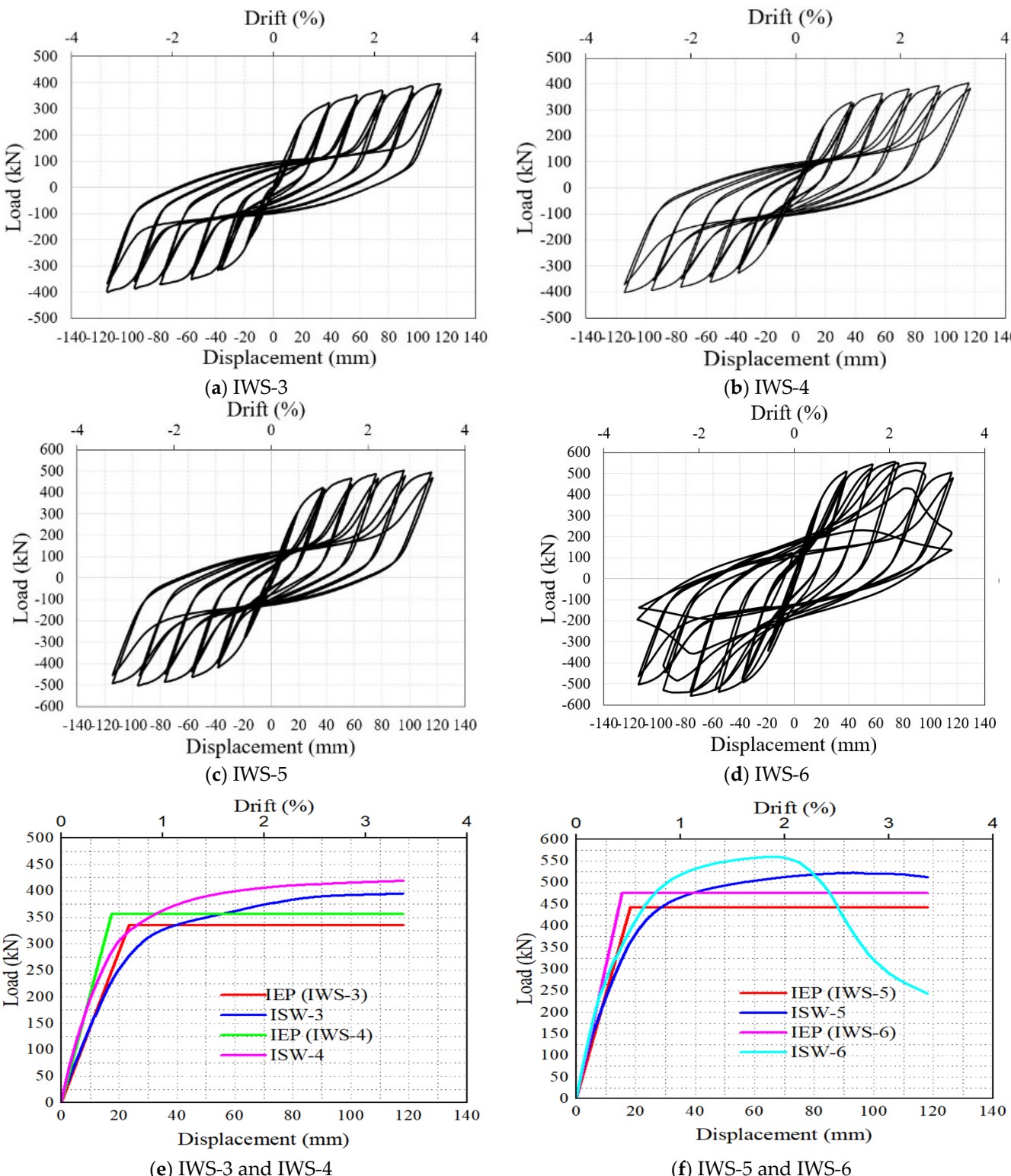

**Figure 19.** Hysteresis, pushover, and IEP response of the FE models.

## 7. Conclusions

In this article, an experimental and numerical study was conducted to investigate the performance of IWS-SPSW and USPSW. Two 1:3 scale three-story SPSW specimens were constructed and examined under cyclic loading. The test findings created and validated each specimen's FE models. The FE models considered material and geometric nonlinearity, excessive deformation, and geometric imperfection. The method of implicit nonlinear solution in FE analysis was used with a quasi-static application. A parametric study was performed to evaluate the effects of the bolt connection of bi-diagonal strips, the yield stress of the infill web strips, and the thickness of the infill web strips. In conclusion, the following key factors are spotlighted:

- The SPSWs specimen showed excellent shear load-bearing, lateral deformation, energy dissipation, and ductility. The ductility characteristics values for USPSW and IWS-SPSW were 4.54 and 5.15, respectively. ISW-SPSW was higher than the USPSW. Additionally, the ISW-SPSW displayed 23.7% less yield and the ultimate load. The USPSW showed load–displacement curve degradation after 2.72 % lateral story drift. At the same story drift and further, till 3.3 %, there was no load–displacement curve degradation of the IWS-SPSW specimen. The energy dissipation of IWS-SPSW was observed higher till 1% story drift than the USPSW. In 2–3.3% story drift, the USPSW showed more excellent energy dissipation because of the first-floor column's inelastic deformation.

- The USPSW infill web plate transferred significant tension stress to the beam columns, which caused the out-of-plane failure of the first-floor columns. The ISW-SPSW columns were stable, and no local or global buckling was observed. This indicates that the infill stirps potentially impact the vertical boundary element's axial force and flexural moments response.

- The load–displacement hysteresis curves and failure mechanisms of SPSWs were compared between the results of finite element models and the experimental results of SPSWs. Because of its ability to accurately identify the load resistance preceding buckling and post-buckling, it was found that the FE model could accurately predict these properties. The FE models accurately modeled post-buckling deformation, pinching phenomenon, and system stiffness. The equivalent plastic strain (PEEQ) and von Mises stress result revealed that FE models could better predict the likelihood of SPSW failure modes and mechanisms.

- The parametric study results indicate that the effects of the bolt connection of bi-diagonal strips, yield stress of the infill web-strips, and infill web-strips thickness were significant in shear load-bearing and out-of-plane buckling of infill web strips. It was found the SW-3, SW-4, and SW-5 models showed good seismic performance. The beam-column connection and length-to-height ratio of the specimen may impact the seismic performance of the proposed IWS-SPSW, which can be studied in future work.

**Author Contributions:** Conceptualization, Z.T.; Formal analysis, Y.H.; Funding acquisition, Z.T.; Investigation, Z.T. and Z.Z.; Methodology, W.A.G.; Project administration, L.W.; Resources, Z.Z.; Software, W.A.G.; Validation, W.A.G.; Writing—original draft, W.A.G.; Writing—review & editing, Y.T. All authors have read and agreed to the published version of the manuscript.

**Funding:** This research is sponsored by the Key Research and Development programs (Key R&D programs) department of science and technology of Yunnan province and the Yunnan Earthquake Engineering Research Institute (YEERI) (grant nos. 202003AC100001).

**Data Availability Statement:** The data supporting this study's findings are available on request from the corresponding author.

**Conflicts of Interest:** The authors declare no conflict of interest.

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
