# Peer review of "Experimental and Numerical Study of an Innovative Infill Web-Strips Steel Plate Shear Wall with Rigid Beam-to-Column Connections"

_buildings, doi:10.3390/buildings12101560_

Round 1
Reviewer 1 Report (New Reviewer)
The Authors done an excellent work on experimental and numerical study of an innovative infill web-strips steel plate shear wall with rigid beam-to-column connections. the below listed suggestions are to improve the quality of he article
1. Below figure 17, the test, should be deleted, mention what result is depicted
2. The paper has several typos. Authors need to proofread the paper to eliminate all of them.
3. The introduction should clearly explain the key limitations of prior work that are relevant to this paper.
4. There are inconsistencies in the notations used through the paper. Please make it consistent.
5. The authors should first give an overview of their solution before explaining the details.
6. Some text must be added to discuss the future work or research opportunities
Author Response
Manuscript ID: buildings-1904459
Title: Experimental and numerical study of an innovative infill web-strips steel plate shear wall with rigid beam-to-column connections
Dear Reviewer #1,
The authors are grateful for the attention that you have paid to read our article. All Your recommendations are reflected in the main text by the YELLOW highlight color. Most of them are highlighted throughout the text. The responses to your comments are presented as follows:
Comment #1: Below figure 17, the test, should be deleted, mention what result is depicted
Answer: Thank you for your comment: We deleted the (Test), and the related result has been modified in the revised manuscript.
Comment #2: The paper has several typos. Authors need to proofread the paper to eliminate all of them.
Answer: Thank you for your deep attention. The typo errors are eliminated; we have carefully revised the paper to improve the grammar and readability.
Comment #3: The introduction should clearly explain the key limitations of prior work that are relevant to this paper.
Answer: Thank you for your valuable comment. It has been modified carefully in the revised manuscript.
Comment #4: There are inconsistencies in the notations used through the paper. Please make it consistent.
Answer: Thank you for your constructive comment. The inconsistency of the paper was checked, and It has been modified in the revised manuscript
Comment #5: The authors should first give an overview of their solution before explaining the details.
Answer: Thank you for your essential comment. This suggestion was carefully considered in the revised manuscript.
Comment #6: Some text must be added to discuss the future work or research opportunities
Answer: Thank you for your attention. In the last part of the conclusion, we have added the future work of this research.
Finally, if any further changes are needed, the authors are willing to fulfill them.

Reviewer 2 Report (New Reviewer)
This manuscript conducted experimental study on the hysteretic behaviour of a type of infill web-strip SPSWs, and compared with the ordinary flat SPSWs. Generally, the test was well introduced. The following questions should be concerned:
1. One of the key problems is that, the advantage of the newly proposed IWS-SPSW compared to the USPSW should be well highlighted, based on the experimental and numerical study. Now I don't see a good explanation on this in the manuscript. Thus, a question arises, why would we use this type of steel plate shear wall?
2. Second, in the conclusion it was stated that, ‘infill web-strips significantly reduced the column's axial force ratio.’ This should be obtained based on the inner force analysis of the boundary column, which however is not seen in the manuscript.
3. section 2.1 is not closely related to the key topic of the manuscript, which is the insight into the mechanism of the IWS-SPSWs.
4. Fig.14 ‘pushover curves of tests’. Actually only cyclic loading tests was conducted, why was there pushover curve from the test?
5. Section 3.5, ‘The specimen Py/Pmax ratios are equivalent to 0.85, indicating high ductility’. How can the factor Py/Pmax evaluate the ductility? Explanation is needed.
6. There are several writing mistakes. For instance, Fig.6 ‘fist floor’?; the last sentence in conclusions is not complete.
7. In the introduction, ‘Researchers have proposed several strategies … caused by diagonal tension in web plates, including…’. Actually, corrugated steel plate shear walls are also new type of SPSWs to solve this problem, which thus should be included in the research review. Consequently, the following important references should be added:
[1] Shear resistance and post-buckling behavior of corrugated panels in steel plate shear walls Thin-Walled Structures. 2018, 131: 816–826.
[2] Ultimate shear resistance and post-ultimate behaviour of double-corrugated-plate shear walls. J Constr Steel Res 2020; 165: 105895.
[3] Shear Resistance and Design of Infill Panels in Corrugated-Plate Shear Walls. Journal of Structural Engineering, 2021, 147(11): 04021179.
[4] Research on the specially-shaped steel corrugated shear walls with horizontal corrugation. J Constr Steel Res 2022; 188: 107012.
Author Response
Manuscript ID: buildings-1904459
Title: Experimental and numerical study of an innovative infill web-strips steel plate shear wall with rigid beam-to-column connections
Dear Reviewer #2,
The authors are grateful for the attention that you have paid to read our article. All Your recommendations are reflected in the main text by the YELLOW highlight color. Most of them are highlighted throughout the text. The responses to your comments are presented as follows:
Comment #1: One of the key problems is that, the advantage of the newly proposed IWS-SPSW compared to the USPSW should be well highlighted, based on the experimental and numerical study. Now I don’t see a good explanation on this in the manuscript. Thus, a question arises, why would we use this type of steel plate shear wall?
Answer: Thank you for your valuable comment. We have highlighted this issue in the revised manuscript in the abstract (“The hysteresis results showed that the IWS-SPSW had high energy dissipation with no severe beam-columns damages; on the other hand, the USPSW displayed severe post-buckling, infill panel cracks and first-floor column damages. Moreover, the IWS-SPSW shear strength did not fall in the test specimen beyond 2.5 % average story drift, where the structure exhibited great seismic behavior. FE models were created and validated with experimental data. It has been proved that the infill web-strips can affect an SPSW system’s high performance and overall energy dissipation.”) in the introduction (“Infill web strips have certain advantages over solid infill web plates, such as reducing the connectivity of web plates to boundary elements; therefore, this can produce less axial force and flexural moment to the boundary elements. Previous cyclic tests [33-35] have shown comparatively large cyclic strain concentrations at the corners where a gap between the horizontal and vertical fin plates caused the corners of the USPSW to fracture. Additionally, unstiffened SPSW ends up remaining relatively thin. The arrangement of large and thin steel plates during construction, particularly the field welding of the thin plates to the boundary columns and beams, can be challenging, and this new system can resolve these problems effectively.”) as well as the performace of IWS- SPSW are explained in the experimental results and conclusion.
Comment #2: Second, in the conclusion it was stated that, ‘infill web-strips significantly reduced the column’s axial force ratio.’ This should be obtained based on the inner force analysis of the boundary column, which however is not seen in the manuscript.
Answer: Thank you for your essential comment. Our conclusion was based on the physical observation of the severe damages to the first-floor USPSW columns, which indicates the infill panel transferred significant stress to the boundary columns, but the IWS-SPSW columns were stable until the end of loading, and no local or global buckling was observed. In order to avoid confusion, we have revised that conclusion in the revised manuscript as follows. (“The USPSW infill web plate transferred significant tension stress to the beam col-umns, which caused the out-of-plane failure of the first-floor columns. The ISW-SPSW columns were stable, and no local or global buckling was observed; this indicates that the infill stirps potentially impact to reduce the vertical boundary elements axial force and flexural moments.”)
Comment #3: Section 2.1 is not closely related to the key topic of the manuscript, which is the insight into the mechanism of the IWS-SPSWs.
Answer: Thank you for your valuable comment. We have eliminated unrelated paragraphs from section 2.1.
Comment #4: Fig.14 ‘pushover curves of tests’. Actually only cyclic loading tests was conducted, why was there pushover curve from the test?
Answer: You are right. We used the wrong term; it should be ( the envelope curve of the specimens), which is obtained by successively connecting the peak points of the hysteresis curve of the first cycle at each stage in the same direction. We have modified it in the revised manuscript.
Comment #5: Section 3.5, ‘The specimen Py/Pmax ratios are equivalent to 0.85, indicating high ductility’. How can the factor Py/Pmax evaluate the ductility? Explanation is needed.
Answer: Thank you for your essential comment. The statement “The specimen Py/Pmax ratios are equivalent to 0.85, indicating high ductility” was wrong. We have modified it in the revised manuscript to “ The specimen Py/Pmax ratios are equivalent to 0.85, suggesting better ductility”. It means the maximum displacement (δmax) of the specimens showing softening behavior was defined as the value corresponding to 0.85 times the maximum load. And the ductility factor is evaluated based on the ( μ=δmax/δy), not by the Py/Pmax ratios.
Comment #6: There are several writing mistakes. For instance, Fig.6 ‘fist floor’?; the last sentence in conclusions is not complete.
Answer: Thank you for your deep attention. The spelling mistakes are fixed, and the last sentence of the conclusion part has been modified in the revised manuscript.
Comment #7: In the introduction, ‘Researchers have proposed several strategies … caused by diagonal tension in web plates, including…’. Actually, corrugated steel plate shear walls are also new type of SPSWs to solve this problem, which thus should be included in the research review. Consequently, the following important references should be added:
[1] Shear resistance and post-buckling behavior of corrugated panels in steel plate shear walls Thin-Walled Structures. 2018, 131: 816–826.
[2] Ultimate shear resistance and post-ultimate behaviour of double-corrugated-plate shear walls. J Constr Steel Res 2020; 165: 105895.
[3] Shear Resistance and Design of Infill Panels in Corrugated-Plate Shear Walls. Journal of Structural Engineering, 2021, 147(11): 04021179.
[4] Research on the specially-shaped steel corrugated shear walls with horizontal corrugation. J Constr Steel Res 2022; 188: 107012.
Answer: Thank you for your constructive suggestion. In the research review, we have included the corrugated steel plate shear walls, and the given references are cited in the revised manuscript.
Finally, if any further changes are needed, the authors are willing to fulfill them.

Reviewer 3 Report (New Reviewer)
Authors have presented experimental and numerical investigations on an innovating steel plate shear walls with moment-resisting boundary frames.
It is an interesting and comprehensive work and of practical importance in the field.
Authors proposed a new type of steel plate shear wall system called infill web-strips (IWS-SPSW) and evaluated its cyclic behavior through experimental and numerical research:
1- An experimental specimen was built and cyclically tested in the lab. Test method and results were adequately discussed in detail.
2- The cyclic performance of the proposed IWS-SPSW system was compared with that of an experimentally tested conventional unstiffened steel plate shear.
3- Analytical finite element models were created and validated against test results. Parametric studies were conducted using the test validated FE models to further investigate the effects of infill-strip bolt connections, web-strips yield strength, and web-strips thickness on the behavior of IWS-SPSW systems.
4- It was demonstrated that the proposed SPSW model with web-strips infill plates can improve the cyclic performance of SPSW systems, while exhibiting a ductile behavior.
Conclusions drawn are appropriate to the objectives of the research work.
Hence paper is worth publishing in this journal. It is recommended for publication.
Round 2
Reviewer 2 Report (New Reviewer)
All the questions from the reviewer have been adderessed, thus the revised manuscript can be accepted.
This manuscript is a resubmission of an earlier submission. The following is a list of the peer review reports and author responses from that submission.
Round 1
Reviewer 1 Report
Please see the attachment.

Author Response
We appreciate the time and efforts the reviewers have dedicated to providing your valuable feedback on our manuscript. We are grateful to the reviewers for their insightful comments on our paper. We have been able to incorporate changes to reflect most of the suggestions provided by the reviewers. We have highlighted the changes within the manuscript.
Here is a point-by-point response to the reviewers’ comments and concerns.
Reviewer 1
Reviewer Comments:
Comment 1: In the “Introduction” section, the third paragraph says: “In order to weaken the web plate, many researchers have proposed different types of SPSWs……”. Explain why researchers need to weaken the web plate.
Author Response: It’s an excellent question. One difficulty in selecting an SPSW system is that the available panel material may be stronger or thicker than needed for a design situation; This will increase the necessary size of horizontal and vertical boundary members (Beam & Columns) as well as foundations demands since these members are generally designed for the strength of the plate; To alleviate this concern, researchers have focused on the use of light gauges, cold-rolled and low-yield strength (LYS) steel for infill panel, and on the placement of a pattern of perforation to decrease the strength and stiffness of the panel. Therefore this issue is discussed as a weakening of the web plate, and the reason and references are provided.
Comment 2: The manuscript lacks the proper literature about SPSWs, the authors shall add a new section “Literature Review”.
Author Response: We tried to bring a summarized, relative, and informative literature review that can provide an integrated literature review to the present research. And it is prevalent in a research article that the literature review should include the introduction section. If the review is not agreed upon, we can add this section separately.
Comment 3: The structure of the “Introduction” section is vague and unorganized, the authors shall explain the problem statement of this research, motivations, research gap, novelty, and implications of this study.
Author Response: We double-checked the (Introduction ) and adequately updated it.
Comment 4: For equations 1 to 8, you should mention the exact location of these equations on the corresponding codes, to make it easy for the reader to find these equations in the codes.
Author Response: We know the readers are researchers and designers; they are familiar with those equations and corresponding codes. And most design provisions codes have one section for the design of this system; we don’t think there should be confusion about it.
Comment 5: Figure 4 has low quality and should be improved, for example, the loading cell and strain gauge are referring to the same picture, also the strong foundation is not clearly shown in the figure. Moreover, explain why you only used one actuator at the top corner of the frame, I mean why you did not use three actuators (one actuator at each floor level).
Author Response: Figure 4 was updated, and we tried to make it more visible.
Comment 6: Poor legends in Fig. 6 and Fig. 7, for example, the reader doesn’t know the location of corner tearing in Fig.6d, I mean which corner is this one, and which floor level, …etc.
Author Response: The figures are updated and made more understandable.
Comment 7: I am not sure the calculated ductility values in section 3.5 are correct, because you did not provide sufficient load vs displacement information after the peak strength. Therefore, in Fig. 8, I would like to see the load vs displacement after load degradation up to 50% of maximum load, this is important to find the ultimate displacement at 80% of maximum load which is usually used to find the ductility factor.
Author Response: To determine the ductility factor from a global force-displacement curve is complicated, and there will be errors. We agree this specimen did not show till the 50% degradation; therefore, we used to determine the ultimate displacement from 85% of the maximum load, which is approximate to the last cycle of the cyclic loading. We updated this adequately in the releted section.
Comment 8: In section 3.2, I think you have typo, do you mean Fig. 7 or 8?
Author Response: Thanks for reminding us; we have fixed this error; it should be (Figure 7)
Comment 9: The quality of technical writing in this manuscript is low, the authors shall improve the writing of this manuscript considering the flow and coherence of discussions.
Author Response: Good suggestion; we revised it carefully and improved the flow and coherence of the paper writing.
Reviewer 2 Report
This paper investigated the cyclic behavior of an innovative infill web-strips steel plate shear wall with rigid beam-to-column connections using both experimental and numerical methods. The topic itself is interesting, and the paper is also well written. I recommend its publication after minor changes. Authors are advised to respond to the comments provided below carefully. The comments are as follows:
(1) The English language is poor, and the current version of the manuscript contains low grammatical errors. The reviewer recommends the authors polish the manuscript.
(2) The authors give a comprehensive literature review on the research on SPSWs. However, most of the cited references are too old. More references in the latest five years should be mentioned to reflect the most recent advancement in this field.You can refer to the following references:
* https://doi.org/10.1016/j.jobe.2020.101821
* https://doi.org/10.1016/j.istruc.2021.06.046
* https://doi.org/10.1016/j.jcsr.2022.107157
* https://doi.org/10.1016/j.jobe.2022.104963
* https://doi.org/10.1007/s13296-021-00529-3
* https://doi.org/10.1016/j.jobe.2021.102844
* https://doi.org/10.12989/scs.2021.39.1.109
(3) In Figure 1, it would have been better to choose the last three floors for the laboratory sample in order to save the effects of axial force and the bending moment of the floors. Are the effects of the upper floors applied in the loading of the laboratory sample?
(4) Fig. 9 needs to be revised. According to Fig.9, the yielding force and displacement is wrong. The two hatched areas should be equal in the schematic figure. But in Fig. 9, the area of the first area is much smaller than the area of the second area for both shapes. The definition of yielding force and displacement is recommended to revise as recommended by the following paper.
** Gorji Azandariani, M.; Gholhaki, M.; Kafi, MA Experimental and numerical investigation of low-yield-strength (LYS) steel plate shear walls under cyclic loading. Engineering Structures 2020, 203.
(5) The energy dissipation is recommended to be plotted versus lateral displacement instead of cycle number.
(6) Similar tests have been done by other researchers on SPSWs under cyclic loading. The authors should clarify what the originality of this research is.
(7) The models of parametric studies should be checked again. Figure 19(d), out-of-plane buckling occurred at the foot of the first-floor column. Also, the rotation has occurred in the gusset plate connecting at the column foot.
(8) In addition to comparing numerical models with experimental results and parametric study, it is recommended that the effects of the type of connections on short, medium, and long floor structures be investigated.
(9) The conclusion section is long. It is better only to present the main results of the research.
Author Response
We appreciate the time and efforts the reviewers have dedicated to providing your valuable feedback on our manuscript. We are grateful to the reviewers for their insightful comments on our paper. We have been able to incorporate changes to reflect most of the suggestions provided by the reviewers. We have highlighted the changes within the manuscript.
Here is a point-by-point response to the reviewers’ comments and concerns.
Reviewer 2
Reviewer Comments:
Comment 1: The English language is poor, and the current version of the manuscript contains low grammatical errors. The reviewer recommends the authors polish the manuscript.
Author Response: we agreed and tried to improve the language errors; we double-checked the manuscript to solve any grammatical problems of the paper.
Comment 2: The authors give a comprehensive literature review on the research on SPSWs. However, most of the cited references are too old. More references in the latest five years should be mentioned to reflect the most recent advancement in this field. You can refer to the following references:
* https://doi.org/10.1016/j.jobe.2020.101821
* https://doi.org/10.1016/j.istruc.2021.06.046
* https://doi.org/10.1016/j.jcsr.2022.107157
* https://doi.org/10.1016/j.jobe.2022.104963
* https://doi.org/10.1007/s13296-021-00529-3
* https://doi.org/10.1016/j.jobe.2021.102844
* https://doi.org/10.12989/scs.2021.39.1.109
Author Response: thanks for this insightful comment. Some of the given references were cited, such as https://doi.org/10.1016/j.jobe.2020.101821, a reference (13) in this article, and others also considered for citations.
Comment 3: In Figure 1, it would have been better to choose the last three floors for the laboratory sample in order to save the effects of axial force and the bending moment of the floors. Are the effects of the upper floors applied in the loading of the laboratory sample?
Author Response: It is a good suggestion. Maybe we can be considered in the future study. For this study, the upper floor load was applied on the top of boundary columns which we discussed in the loading section.
Comment 4 : Fig. 9 needs to be revised. According to Fig.9, the yielding force and displacement is wrong. The two hatched areas should be equal in the schematic figure. But in Fig. 9, the area of the first area is much smaller than the area of the second area for both shapes. The definition of yielding force and displacement is recommended to revise as recommended by the following paper.
** Gorji Azandariani, M.; Gholhaki, M.; Kafi, MA Experimental and numerical investigation of low-yield-strength (LYS) steel plate shear walls under cyclic loading. Engineering Structures 2020, 203.
Author Response: Figure 9 is revised based on the recommended article.
Comment 5: The energy dissipation is recommended to be plotted versus lateral displacement instead of cycle number.
Author Response: a good recommendation, but with the method that we calculated the energy dissipation, it is better to make the graph logical and straightforward.
Comment 6: Similar tests have been done by other researchers on SPSWs under cyclic loading. The authors should clarify what the originality of this research is.
Author Response: We agree there is similar research on the SPSW, But this paper presents an innovative unstiffened steel shear wall that can solve the several problems that came from a conventional SPSW; this issue was discussed in the Introduction section paragraphs 2-3. “The fundamental challenge of establishing the diagonal tension field action on the web plate in these types of walls is to resist lateral load [2,3]. So far, the diagonal tension field has caused boundary elements to experience critical axial forces and flexural moments [7,8]. The design of columns with significant internal forces, particularly in multistory constructions, is complex. The design force based on the seismic design provisions of the code [1] is used to establish the web plate thickness. Then, because of the diagonal tension field in the web plate, the boundary elements are intended to be elastic under maximum stress [11]. Moreover, the thicknesses of solid web panels are generally impacted by non-resistance control elements such as installation, application, and welding circumstances, resulting in a thicker web panel, which facilitates the transmission of force to the boundary components [9-17]. In order to weaken the web plate, many researchers have proposed different types of SPSWs, including SPSWs light gauge SPSWs [18,19], low yield point SPSWs[20-22], SPSWs with slit [23,24], SPSWs with a partially connected web plate [25,26], and self-centering steel plate shear wall with infill web-strips and solid web plates [27,28].
This paper presents an innovative infill web-strips steel plate shear walls (IWS-SPSW) system. This system is composed of horizontal and vertical boundary elements, and the infill web strips are arranged uniformly to a condition in which the tension field's inclination angle is adjusted. A fin plate connects the strips to the beam and column elements. The wider length bi-diagonal strips are restrained together by a bolt connection to avoid significant out-of-plane deformation. Infill web strips have certain advantages over solid infill web plates, such as reducing the connectivity of web plates to boundary elements; therefore, this can produce less axial force and flexural moment to the boundary elements. Previous cyclic tests [2,3,8] have shown comparatively large cyclic strain concentrations at the corners where a gap between the horizontal and vertical fin plates caused the corners of the USPSW to fracture.
Additionally, unstiffened SPSW ends up remaining relatively thin. The arrangement of large and thin steel plates during construction, particularly the field welding of the thin plates to the boundary columns and beams, can be challenging, and this new system can resolve these problems effectively. Based on the novelty of the IWS-SPSW system, this study aims to examine the mechanical properties of this proposed shear wall under a cyclic lateral loading test and numerical analysis.”
Comment 7: The models of parametric studies should be checked again. Figure 19(d), out-of-plane buckling occurred at the foot of the first-floor column. Also, the rotation has occurred in the gusset plate connecting at the column foot.
Author Response: The models of parametric studies were rechecked, and the errors were fixed.
Comment 8: In addition to comparing numerical models with experimental results and parametric study, it is recommended that the effects of the type of connections on short, medium, and long floor structures be investigated.
Author Response: it’s a very nice comment; we are preparing a new prototype test for the different types of connections, but in this study, we think it will be a lengthy article; therefore, we agree to consider the type of connections, the height of the SPSW and the length of the SPSWs parameters in the future work.
Comment 9: The conclusion section is long. It is better only to present the main results of the research.
Author Response: We revised the conclusion and tried to make as much as short and precise.
Reviewer 3 Report
The submitted Article buildings-1835369 entitled: “Experimental and numerical study of an innovative infill web-strips steel plate shear wall with rigid beam-to-column connections” is an experimental and numerical study. The authors have proposed a new infill web-strips steel plate shear walls (IWS-SPSW) system which was tested under cyclic lateral loading. The authors have additionally performed finite element parametric analysis to further investigate the mechanical performance of the materials used in the proposed system.
The authors have performed a very interesting study with a well-planned experimental work and a very detailed numerical analysis. The manuscript is easy to follow, the figures are of good quality and the results are very interesting. Congratulations on the very good research.
Author Response
We appreciate the time and efforts the reviewers have dedicated to providing your valuable feedback on our manuscript. We are grateful to the reviewers for their insightful comments on our paper.
Round 2
Reviewer 1 Report
The required comments were not addressed professionally. Most of the authors' replies do not answer the suggested comments.
Reviewer 2 Report
be accepted.